

# Drop it all: extraction-free detection of targeted marine species through optimized direct droplet digital PCR

Michelle Scriver[1,2], Ulla von Ammon[1], Cody Youngbull[3], Xavier Pochon[1,2], Jo-Ann L. Stanton[4], Neil J. Gemmell[4] and Anastasija Zaiko[1,5]

[1] Biosecurity Group, Cawthron Institute, Nelson, New Zealand
[2] Institute of Marine Science, University of Auckland, Auckland, New Zealand
[3] Nucleic Sensing Systems, LCC, Saint Paul, Minnesota, United States
[4] Department of Anatomy, School of Biomedical Sciences, University of Otago, Dunedin, New Zealand
[5] Sequench Ltd, Nelson, New Zealand

Corresponding author
Michelle Scriver,
mscr601@aucklanduni.ac.nz

## ABSTRACT

Molecular biomonitoring programs increasingly use environmental DNA (eDNA) for detecting targeted species such as marine non-indigenous species (NIS) or endangered species. However, the current molecular detection workflow is cumbersome and time-demanding, and thereby can hinder management efforts and restrict the "opportunity window" for rapid management responses. Here, we describe a direct droplet digital PCR (direct-ddPCR) approach to detect species-specific free-floating extra-cellular eDNA (free-eDNA) signals, *i.e.*, detection of species-specific eDNA without the need for filtration or DNA extraction, with seawater samples. This first proof-of-concept aquarium study was conducted with three distinct marine species: the Mediterranean fanworm *Sabella spallanzanii*, the ascidian clubbed tunicate *Styela clava*, and the brown bryozoan *Bugula neritina* to evaluate the detectability of free-eDNA in seawater. The detectability of targeted free-eDNA was assessed by directly analysing aquarium marine water samples using an optimized species-specific ddPCR assay. The results demonstrated the consistent detection of *S. spallanzanii* and *B. neritina* free-eDNA when these organisms were present in high abundance. Once organisms were removed, the free-eDNA signal exponentially declined, noting that free-eDNA persisted between 24–72 h. Results indicate that organism biomass, specimen characteristics (*e.g.*, stress and viability), and species-specific biological differences may influence free-eDNA detectability. This study represents the first step in assessing the feasibility of direct-ddPCR technology for the detection of marine species. Our results provide information that could aid in the development of new technology, such as a field development of ddPCR systems, which could allow for automated continuous monitoring of targeted marine species, enabling point-of-need detection and rapid management responses.

## INTRODUCTION

Biomonitoring practitioners often face the daunting task of continuously monitoring targeted species, such as bioindicators and endangered species, and surveillance for biological threats, such as non-indigenous species (NIS), in vast and complex marine environments. Effective biomonitoring requires regular monitoring to assess the impact of management practices (*Gold et al., 2021*). Swift and accurate detection of rare species like marine NIS and endangered species, followed by rapid responses upon detection, is essential to maximize the effectiveness of management efforts (*McDonald et al., 2020*; *Meyerson & Reaser, 2002*; *Vander Zanden et al., 2010*; *Wittenberg & Cock, 2001*; *Xia et al., 2021*). Therefore, molecular detection technology such as environmental DNA (eDNA), which is a non-invasive, cost-effective, sensitive, and rapid technique, is gaining much interest for integration into biomonitoring and biosecurity surveillance programs (*Borrell et al., 2017*; *De Brauwer et al., 2023*; *Duarte et al., 2021*; *Larson et al., 2020*; *Pearman et al., 2021*; *Zaiko et al., 2018*).

To aid in the uptake of eDNA tools in routine biomonitoring, recent studies have begun to optimize and standardize eDNA workflows and methodology (*De Brauwer et al., 2023*; *Fernandez et al., 2021*; *Jeunen et al., 2019*; *Zaiko et al., 2022*). However, the current molecular detection workflows are still quite cumbersome, require access to specialized laboratory facilities and expertise in sample processing, and often involve complex sample collection and logistics (*Bowers et al., 2021*; *Jeunen et al., 2022*; *Larson et al., 2020*; *Thomas et al., 2020*). These challenges can make sample collection arduous and introduce errors and delays, hindering management efforts and limiting the window of opportunity for rapid management responses and adjustments (*Ponce, Arismendi & Thomas, 2021*).

Direct PCR amplification could offer a promising solution, by allowing the addition of the sample to the PCR reaction without the need for prior sample preservation, DNA extraction, purification, or quantification; thus, bypassing traditional sample manipulation (*Cascella et al., 2015*; *Cavanaugh & Bathrick, 2018*). This technique has been successfully used for bacterial detection in clinical trials (*Mora, Abdel-Kader & Maklad, 2013*; *Nakao & Popovic, 1997*), environmental (*Benson, Fode-Vaughan & Collins, 2004*) and mixed samples (*Pacocha et al., 2019*), as well as touch DNA in forensic science (*Cavanaugh & Bathrick, 2018*), DNA barcoding of macroinvertebrate tissues (*Wong et al., 2014*), metabarcoding marine bacterial communities (*Stojan et al., 2023*) and species identification in wildlife forensics (*Kitpipit, Thanakiatkrai & Chotigeat, 2013*).

The complex nature of eDNA, which consists of a mixture of genetic material from living organisms, expelled cells and particles, extracellular DNA bound to substrates, and free-floating eDNA, makes it an ideal target for direct PCR (*Barnes & Turner, 2015*; *Pawlowski, Apotheloz-Perret-Gentil & Altermatt, 2020*; *Zaiko et al., 2022*). Direct PCR is particularly suitable for detecting specific states of eDNA, such as free-floating extracellular eDNA (free-eDNA), which is not bound to other particles or within cells and can originate from cellular debris or damaged cells that can be easily lysed with high temperatures. Furthermore, the combination of direct PCR with droplet digital PCR (ddPCR) technology allows for the detection of trace amounts of free-DNA while minimizing errors,

contamination, time, and cost (*Cao, Griffith & Weisberg, 2016*; *Capo et al., 2021*; *Templeton et al., 2015*). Additionally, ddPCR assays also present advantages over conventional quantitative PCR (qPCR) methods, and demonstrated increased accuracy, sensitivity, and effectiveness in eDNA studies for targeting eDNA concentrations in water (*Cao, Griffith & Weisberg, 2016*, *Doi et al., 2015*; *Nathan et al., 2014*; *Wood et al., 2019*). Therefore, by combining the absolute quantification capability of ddPCR with the advantages of direct PCR, direct-ddPCR has the potential to enable detection directly from water samples, simplifying the workflow and facilitating response and management programs. Despite the tremendous potential and advantages of direct-ddPCR technology, its application for the direct detection of free-eDNA in seawater remains unexplored, offering an opportunity for further research and development.

This study investigates the feasibility of detecting species-targeted free-eDNA from saltwater by analyzing seawater samples using direct-ddPCR, bypassing the filtration and DNA extraction steps. To achieve this goal, the initial focus was on optimizing direct-ddPCR assays to reduce inhibition caused by salt. Although ddPCR has shown better performance than qPCR for amplifying low eDNA levels in the presence of inhibitors, salt can still affect the reaction (*Davalieva & Efremov, 2010*; *Mauvisseau et al., 2019a*; *Sedlak, Kuypers & Jerome, 2014*). Once an optimized protocol was established using free-eDNA from preserved samples, a proof-of-concept aquarium experiment was designed with the presence of three known marine organisms, *Sabella spallanzanii*, *Bugula neritina* and *Styela clava*. In this instance, these marine NIS were chosen as targeted organisms, given their requirement for regular monitoring and swift responses and therefore their detection would greatly benefit from direct-ddPCR technology. The objectives of the present study were: (i) to explore the immediate detection of targeted marine species from free-eDNA in seawater samples using direct-ddPCR technology, (ii) to determine the influence of species characteristics and biomass on the detection of free-eDNA, and (iii) to assess the longevity of free-eDNA signal in the system once the organisms are removed.

## MATERIALS AND METHODS

Portions of this text were previously published as part of a preprint (https://doi.org/10.22541/au.169001354.48733131/v1).

### *In-vitro* optimization trials for free-floating environmental DNA detection in seawater

Individual Nalgene™ square polycarbonate bottles (Thermo Fisher Scientific, Waltham, MA, USA) were filled with 250 mL of one of the following: tap water, purified water (Milli-Q®; Millipore Sigma™, Burlington, MA, USA), locally collected seawater or artificial seawater with varying salinities (Red Sea Salt-Copepod salt; Red Sea, Germany). New Zealand marine NIS organisms used for testing included *Sabella spallanzanii* (Gmelin, 1791)—a large Mediterranean fanworm, *Styela clava* (Herdman, 1881)—a leathery club tunicate and *Bugula neritina* (Linnaeus, 1758)—a bush-like, calcified bryozoan, all of which were preserved in 99% ethanol and placed individually in separate
NalgeneᵀᴹM bottles. To simulate eDNA release, the organisms were vigorously shaken within the bottles for several minutes. Subsequently, water aliquots were collected from each bottle and directly added to species-specific ddPCR reactions.

The direct-ddPCR assays were optimized to minimize salt inhibition, and various additives and assay manipulations were tested. This included evaluating PCR additives such as glycerol, dimethyl sulfoxide (DMSO), Tween-20, and bovine serum albumin (BSA), as well as the assessment of sample volume, pH adjustment buffers and bases (1M potassium hydroxide (KOH), 1X Tris-acetate-EDTA (TAE), 1M tris aminomethane (Tris) pH 8.0, and 1M 4-(2-hydroxyethyl)-1-piperazineethanesulfonic acid (HEPES), pH 7.2), PCR annealing temperature gradient (ranging from 54 °C to 62 °C), and the adjustment of primer and probe concentrations. Additionally, different ddPCR master mixes, including ddPCRᵀᴹ Multiplex Supermix (No dUTP) (BioRad, Hercules, CA, USA) and ddPCRᵀᴹ Supermix for Residual DNA Quantification (BioRad, Hercules, CA, USA), were compared. Detailed information regarding the testing and optimization procedures can be found in the Supplemental Material (File S2).

## Experimental setup and organism collection

An aquarium-based experiment was conducted between January and February 2023 at a Physical Containment Level 2 (PC2) facility. All work was done in a controlled laboratory setting, and in accordance with biosecurity regulations, *i.e.*, permission to handle organisms under sections 52 and 53 of the Biosecurity Act 1993 (Notice No. MPI 111).

Seven transparent polycarbonate 26 L tanks with lids and individual aeration pumps were enclosed within a temperature-controlled room (21.3 °C). Experimental tanks were maintained without water exchange to eliminate the effects of inflow on eDNA concentrations and water mixing between tanks. Each tank was filled with UV-treated seawater with a pH between 8–8.1, salinity = 35 ppt, and water temperature = 20 °C. Prior to the start of the experiment, the tanks and the system underwent a cleaning process according to facility procedures: scrubbing the tanks and recirculating the system with a 200-ppm bleach solution for two days. Subsequently, the system was rinsed with sodium thiosulfate to neutralize any remaining chlorine and thoroughly rinsed with distilled water.

The aquarium experiment was set up with three morphologically distinct marine invasive species: *S. spallanzanii*, *S. clava* and *B. neritina*. *Sabella spallanzanii* individuals were collected from a pontoon in Marsden Cove Marina (Marsden Bay, New Zealand: 35.84°S, 174.47°E; 16 January 2023) with local seawater and were shipped in containers on ice to the laboratory. Both *S. clava* and *B. neritina* used in the aquarium experiment were sourced from the side of pontoons from the local Nelson Marina (Nelson, New Zealand: 41.26°S, 173.28°E; 17 January 2023) and immediately transported to the laboratory.

The collected specimens were left to acclimate at the laboratory in tanks with seawater for 5 days without feeding. At the beginning of the experiment (designated as time point zero), the live specimens were distributed among three experimental tanks, with an additional fourth tank left empty as a negative control. The allocation of specimens to each tank was determined based on their size and the number of specimens (this factor is referred to as "biomass or biomass treatment" hereafter, Table 1). The total species weight

**Table 1 Biomass of non-indigenous species used in aquarium experiment.**

| Tank | Species | Total weight of species in the tank (g) | Number of organism in the tank | Referred biomass |
|---|---|---|---|---|
| 1 | *Sabella spallanzanii* | 19.8 | 2 | Medium |
| | *Styela clava* | 9.6 | 1 | High |
| | *Bugula neritina* | 4.7 | 2 | High |
| 2 | *Sabella spallanzanii* | 32.7 | 3 | High |
| | *Styela clava* | 7.6 | 1 | Medium |
| | *Bugula neritina* | 2.2 | 1 | Medium |
| 3 | *Sabella spallanzanii* | 6.3 | 1 | Low |
| | *Styela clava* | 4.5 | 1 | Low |
| | *Bugula neritina* | 0.4 | 1 | Low |

Note:
The biomass of the *Sabella spallanzanii, Bugula neritina* and *Styela clava* grouped by tank used in the aquarium experiment. The total biomass of each species was determined as the weight of individuals within the respective tank. Depending on the total weight within a tank, each species was assigned a classification of high, medium, or low biomass based on relative weight within species.

per tank for each species was determined by weighing the organisms after their removal. Note that the assigned biomass treatments (high, medium, and low) were based on the relative weight within each species, rather than between species. To ensure that *S. spallanzanii* and *B. neritina* stood upright, organisms were attached to a sterilized stainless-steel bolt using plastic ties (Fig. S1). *Styela clava* had a string attached to its stem to hang vertically down, to mime their usual orientation in the environment (Fig. S1).

## Sample collection

Sterile dual filter T.I.P.S® PCR 50–1,000 µL Tip (76 mm) (Eppendorf, Hamburg, Germany) on a compatible micropipette were used to collect water samples (1 mL) from each of the four tanks, 5–7 cm below the surface (Fig. 1). Samples were collected immediately after adding the organisms and then at 4, 8, 24, 48, 72, 96, and 192 (8 days) h. At each sampling occasion, six replicates were taken randomly, targeting different locations in the tank (*i.e.*, back left, back right, front left, front right, and two samples from the middle). Samples were collected in microcentrifuge tubes and kept on ice until further processing (c. 1 h).

After removing the organisms, samples were collected immediately and then at 4, 8, 24, 48, and 72 h as described above. Removed organisms were weighed and photographed (Fig. S2).

## Direct droplet digital polymerase chain reaction

Direct droplet digital PCR (direct-ddPCR) was conducted in an automated droplet generator (QX200™ Droplet Digital PCR System; BioRad, Hercules, CA, USA). Copy numbers (per µL) of the *Cytochrome c oxidase* subunit 1 (COI) gene were measured in all samples using primers and probes specific to *S. spallanzanii* (*Wood et al., 2017*) and *S. clava* (*Gillum, 2014*) and primers specific to *B. neritina* (*Kim et al., 2018*) (Table 2). To compare the results between the two ddPCR chemistries, both the hydrolysis probe (TaqMan, Carlsbad, CA, USA) and DNA binding dye (EvaGreen®, Taoyuan City, Taiwan) assays were utilized in this study. The *S. spallanzanii* and *S. clava* direct-ddPCR

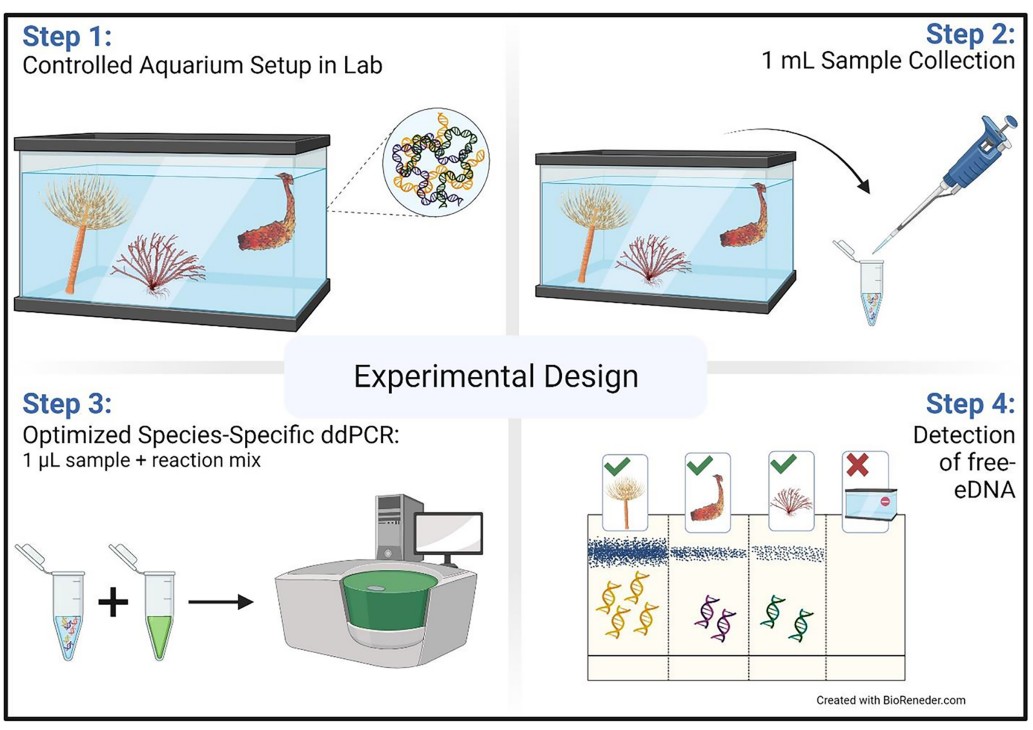

**Figure 1 Aquarium experimental design.** Schematic representation of the aquarium experiment. The schematic shows the four-step sampling procedure carried out for the detection of free-floating extra-cellular environmental DNA (free-eDNA) with direct droplet digital polymerase chain reaction (direct-ddPCR). This figure was made in © BioRender: biorender.com (CC-BY-NC-ND).

**Table 2 Species-specific primers and probes.**

| Target species/Region | Reference | Target size (bp) | Primers & probe | Sequence |
|---|---|---|---|---|
| *S. spallanzanii* | *Wood et al. (2017)* | 90 | Sab3-qPCR-F | 5′-GCTCTTATTAGGCTCTGTGTTTG-3′ |
| | | | Sab3-qPCR-R | 5′-CCTCTATGTCCAACTCCTCTTG-3′ |
| | | | Sab3-qPCR-Probe | 5′-FAM/AAATAGTTCATCCCGTCCCTGCCC/BkFQ-3′ |
| *S. clava* | *Gillum (2014)* | 150 | SC1F | 5′-TCCGGCGGTAGTCCTTTTATT-3′ |
| | | | SC1R | 5′-GAGATCCCCGCCAAATGTAA-3′ |
| | | | SC1-Probe | 5′-HEX/TTAGCTAGGAACTTGGCCCA/NFQ-3′ |
| *B. neritina* | *Kim et al. (2018)* | 185 | BuNe_SF | 5′-GGTACATTATACTTTTTATTTGGAC-3′ |
| | | | BuNe_SR | 5′-CCCCCA ATTATAACTGGTATG-3′ |

**Note:**
Species-specific primers and probes used in direct droplet digital polymerase chain reaction (direct-ddPCR) assay to the target marine non-indigenous species; *Sabella spallanzanii, Bugula neritina* and *Styela clava*.

assays were performed in duplex, as the hydrolysis Sab3-qPCR-Probe was dual-labelled with a 5′ 6-carboxyfluorescein (6-FAM) fluorescent tag and a 3′ Black Hole Quencher. In contrast, the hydrolysis SC1-Probe was designed with a 5′ hexachlorofluorescein (HEX) fluorescent tag and a 3′ non-fluorescent quencher (NFQ). The duplex direct-ddPCR reaction included 10 µL of 2X ddPCR™ Supermix for Probes (No dUTP) (BioRad, Hercules, CA, USA), 1 µL of each primer and probe at 10 pmol, 1 µL of the collected water

sample and 4 µL of sterile water for a total volume of 21 µL. For the *B. neritina* direct-ddPCR assay, each direct-ddPCR reaction included 10 µL of 2X QX200™ ddPCR™ EvaGreen Supermix (BioRad, Hercules, CA, USA), 0.5 µL of each primer at 10 pmol, 1 µL of the collected water sample and 9 µL of sterile water for a total volume of 21 µL. The BioRad QX200™ droplet generator partitioned each reaction mixture into nanodroplets by combining 20 µL of the reaction mixture with 70 µL of BioRad droplet oil, either for probes (Automated Droplet Generation Oil for Probes; BioRad, Hercules, CA, USA) or for EvaGreen (Automated Droplet Generation Oil for EvaGreen; BioRad, Hercules, CA, USA).

The duplex, *S. spallanzanii* and *S. clava* direct-ddPCR assay used the following cycle conditions: hold at 95 °C for 10 min, 40 cycles of 95 °C for 30 s, 57 °C 1 min, and a final enzyme deactivation step at 98 °C for 10 min. The *B. neritina* direct-ddPCR assay used the following cycle conditions: hold at 95 °C for 10 min, 40 cycles of 95 °C for 30 s, 57 °C 1 min, and a final signal stabilization and enzyme deactivation step at 4 °C for 5 min and 90 °C for 5 min. The plates were then analyzed on the QX200™ instrument, including at least one negative control (RNA/DNA-free water Life Technologies) and one positive control (genomic DNA extracted from *S. clava* and *S. spallanzanii* or *B. neritina)*. Based on our experience and observation of ddPCR noise (*e.g.*, proportions of fluorescing droplets in water blanks), the detection for all assays was set above the maximum value of the negative controls in the experiment, *i.e.*, 0.08 copies/µL for the *B. neritina* and *S. clava* direct-ddPCR assays and 0.130 copies/µL for the *S. spallanzanii* direct-ddPCR assay.

## Limit of detection and limit of quantification assay

To determine the limit of detection (LOD) and quantification (LOQ) for the direct-ddPCR of two assays, a serial dilution of the positive control (genomic DNA extracted from either *S. clava* and *S. spallanzanii* or *B. neritina* (20 ng/µL)) was performed and analyzed. Genomic DNA was initially diluted to 200 pg (1/100 dilution). Subsequently, a 10-fold series of 2X dilutions (*e.g.*, from 1/100 to 1/102,400) was performed, resulting in a final concentration of 0.195 pg for both assays. Six replicates of each dilution and negative control were included in both series, and all dilutions were performed with fresh seawater from the aquarium experiment.

## Data analysis

All statistical analyses and visualizations were conducted in R version 4.2.1 software (*R Core Team, 2023*). A Kruskal-Wallis test and Wilcoxon rank sum test were performed to determine whether there was a significant difference in free-eDNA signal for each species between tanks and between species. To analyze the detection of free-eDNA following the removal of organisms, an exponential decay model was fitted using the 'easynls' package in the R software (*Kaps & Lamberson, 2009*; *R Core Team, 2023*; *Wood et al., 2020*). Calculations for limit of detection (LOD), for each direct-ddPCR assay, were based on 95% confidence limit where the lowest level of detection was greater than the maximum value of negative controls (*Baker et al., 2018*). To infer the limit of quantification (LOQ) of the direct-ddPCR assays, the coefficient of variation (CV) was calculated for each standard, and the LOQ was defined as lowest standard concentration with a CV value below 35%

(*Klymus et al., 2020*). Evaluation of the linearity of quantitative measurement (quantitative linearity) for the direct-ddPCR assays was assessed by the log10-transformed copy concentration measured by direct-ddPCR plotted against the log10-transformed input ng of DNA and fitting it with a linear regression (*Zhao et al., 2016*). To determine relationships between biomass (weight in g) and free-eDNA concentrations determined by direct-ddPCR, the "lme4" package in R was used to fit a generalized linear mixed model (GLMM). The mixed model was used to consider factors such as species, tank, and weight as predictors of free-eDNA concentration and assess their effects on the variability in free-eDNA concentrations.

## RESULTS

### Pilot testing and assay optimization for free-floating environmental DNA detection in seawater

Despite trying several methods, none of the additives, buffers, or pH adjustments used could completely eliminate the inhibition of saltwater (File S2). To mitigate the inhibition in observed gene copy numbers due to salt, sample dilution was found to be necessary to detect free-eDNA (Fig. S3). The best volume proportions for the direct-ddPCR assays were determined to be 20 μL of ddPCR master mix (Supermix, primers/probes and water) and 1 μL of the seawater sample (Fig. S3). Additionally, the primer concentration for the *B. neritina* assay was modified to 250 nM instead of 450 nM. An annealing temperature of 57 °C was established as optimal for all assays following the temperature gradient analysis.

### Detection of free-floating environmental DNA from aquaria with organisms present

There was no amplification in samples collected from the negative control tank or the no-template direct-ddPCR controls throughout the entire experiment.

The amplitude of detection from free-eDNA varied and depended on the biomass treatment and/or targeted species. For *S. clava*, low to no detectable free-eDNA concentrations were observed at most time points when organisms were present, regardless of the tank (Fig. 2A). Conversely, *B. neritina* consistently exhibited free-eDNA detection throughout all time points, but only in the high biomass treatment (Fig. 2B). On the other hand, *S. spallanzanii* consistently showed free-eDNA detection at all time points and in all tanks (Fig. 2C). The overall highest free-eDNA copy numbers (13.3 copies/μL) were observed in the *S. spallanzanii* tank with the highest fanworm biomass (three organisms and a total weight of 32.7 g), at the 192 h sampling time point (Fig. 2 and Table S1).

The free-eDNA concentrations among the three species were significantly different (Kruskal-Wallis chi-squared = 58.152, df = 2, $p$-value < 0.0001 ($p$ = 2.358e−13)). Further pairwise comparisons revealed significant differences between *S. spallanzanii* and *S. clava* ($p$ = 0.0037), as well as between *S. spallanzanii* and *B. neritina* ($p$-value < 0.0001 ($p$ = 9.3e−13)), but there was no significant difference between *S. clava* and *B. neritina* ($p$ = 0.8207). It is important to note that *S. clava* free-eDNA had a much lower positive

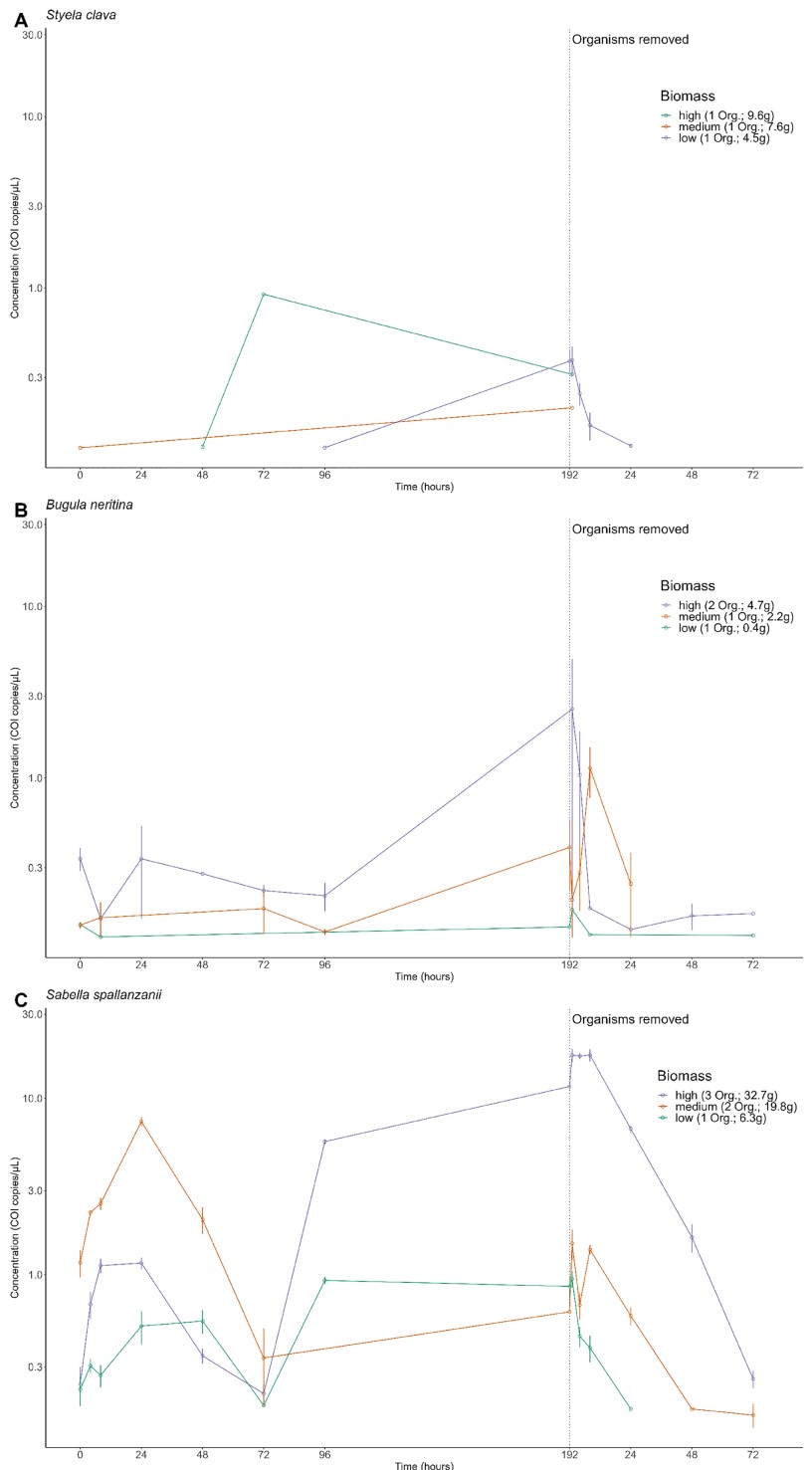

**Figure 2 Extracellular free floating environmental DNA detected for non-indigenous species by tank.** Concentration (copies/μL) of the *Cytochrome c oxidase subunit 1* (COI) gene present in the aquarium grouped by each tank containing different organism's biomass, referring to the number of organisms (Org.) and total weight of the species in grams, (high, medium, or low) for *Styela clava* (A), *Bugula neritina* (B) *and Sabella spallanzanii* (C). Note the biomass categories represent a classification system based on the number of organism and the total weight, in grams, of each species within the tank.

**Figure 2** (continued)
The dotted line indicates the time of organism removal from the aquarium and sample collection after organism removal resets at time point 1 h. Y-axis is presented on a logarithmic scale.

detection rate (6.25%) compared to *B. neritina* (24.3%) and *S. spallanzanii* (74.3%) (Fig. 2 and Table S1).

There were significant differences in *S. spallanzanii* free-eDNA copies/µL between high and low biomass treatments ($p = 0.0015$) as well as low and medium biomass treatments ($p < 0.0001$ ($p = 5.4e-09$)), but not between high and medium biomass treatments ($p = 0.258$) (Kruskal-Wallis chi-squared = 29.715, df = 2, $p$-value $< 0.0001$ ($p = 3.528e-07$)) (Fig. 2C and Table S2). Similarly, for *B. neritina*, significant differences were between high and low biomass treatments ($p = 0.043$) and medium and high biomass treatments ($p = 0.042$), but not between low and medium biomass treatments ($p = 1.0$) (Kruskal-Wallis chi-squared = 8.922, df = 2, $p$-value = 0.011) (Fig. 2B and Table S2). In contrast, no significant differences were observed among the three biomass treatments for *S. clava* (Kruskal-Wallis chi-squared = 2.5247, df = 2, $p$-value = 0.283) (Fig. 2A and Table S2).

## Detection of free-floating environmental DNA after removal of organisms

After organisms were removed at the 192 h time point, an exponential decrease in free-eDNA concentrations was observed. For *S. clava* in the low biomass treatment, the free-eDNA signal was undetectable around 24 h after organism removal (Fig. 3A). However, for *B. neritina* and *S. spallanzanii*, some free-eDNA signal was still detectable for up to 72 h, in the medium and low biomass treatments, and the high and medium biomass treatments, respectively (Figs. 2B and 2C).

Notably, for *S. clava*, no detection of free-eDNA was observed in the high or medium biomass treatments after the removal of organisms (below 0.08 copies/µL after time point zero) (Fig. 2A). In the case of *B. neritina*, only one out of the six replicates in the high and low biomass treatments showed detectable levels of free-eDNA at 72 h (0.162 and 0.120 copies/µL, respectively). In contrast, all six replicates in the high biomass treatment for *S. spallanzanii* exhibited detectable levels of free-eDNA at 72 h (average = 0.257 copies/µL), and two replicates in the medium biomass treatment also showed detectable levels (average = 0.160 copies/µL).

For *S. spallanzanii*, the free-eDNA concentrations remained relatively stable within the first 8 h in the high biomass treatment, averaging 17.5 copies/µL (STDEV = 2.82). However, a decline was observed at the 24 h mark (average = 6.72 copies/µL). The exponential signal decrease continued after 24 h, and at the 72 h mark, the average free-eDNA concentration decreased to an average of 0.257 copies/µL (Figs. 2C and 3B).

The goodness of fit for the exponential models was evaluated using the R-squared values. *Sabella spallanzanii* free-eDNA showed the best fit with an R-squared value of 0.9735 (Fig. 3B), while *S. clava* (Fig. 3A) and *B. neritina* (Fig. 3C) had R-squared values of 0.8225 and 0.8545, respectively (Table S3).

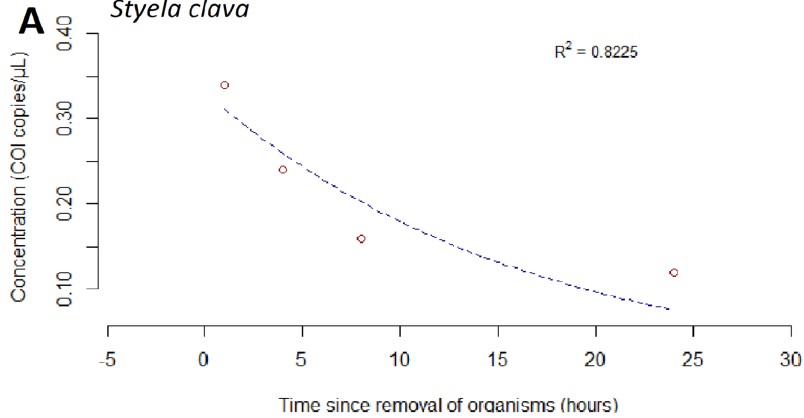

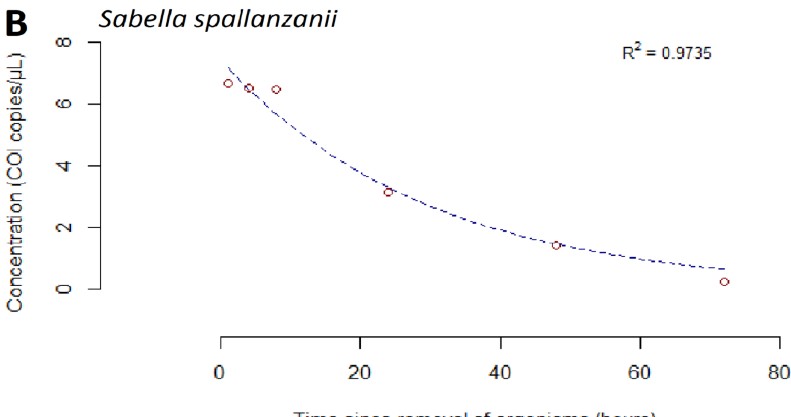

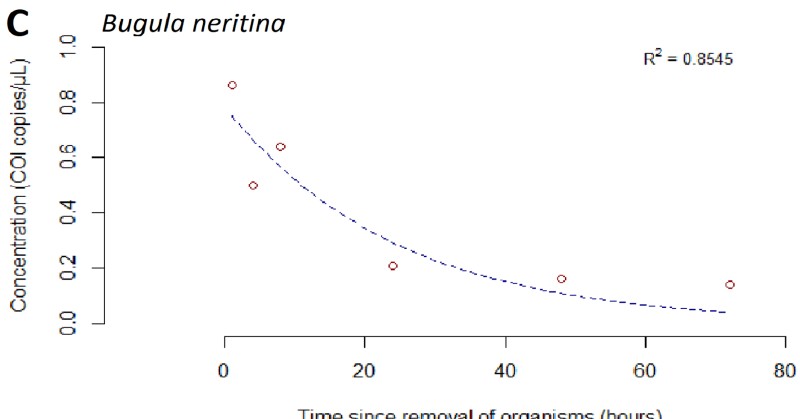

**Figure 3 Exponential model for detected extracellular free floating environmental DNA for non-indigenous species after organism removal.** Time-dependent changes in average environmental DNA copies/μL (based on detection of the *Cytochrome c oxidase subunit 1* (COI) gene) for *Styela clava* (A), *Sabella spallanzanii* (B) and *Bugula neritina* (C) after organism removal. Data for each species across all three tanks were averaged and exponential model, $y = ae^{-bx}$, was applied to the raw data. $R^2$ values indicate the closeness of the fit of raw data to the fitted exponential model. Note different y-axis scales.

**Table 3 Limit of detection and limit of quantification for DNA in seawater by target species.**

| Target species | LOD (copies/µL) | LOQ (copies/µL) |
|---|---|---|
| *Styela clava* | 0.234 | 24.000 |
| *Sabella spallanzanii* | 0.380 | 0.698 |
| *Bugula neritina* | 0.536 | 0.739 |

### Limits of detection and quantification

In general, the dilution series showed an exponential decrease in DNA concentrations for all three species and direct-ddPCR reaction exhibited good linearity (all $R^2 > 0.737$, $p$-value < 0.0001) (Fig. S4). However, for *S. clava*, the DNA concentrations exhibited stability across dilutions of 1/1,600 and 1/6,400, showing a consistent mean concentration of 0.232 copies/µL (STDEV = 0.020). The lower 95% confidence limit ranged from 0.075 to 0.128, which falls just outside the mean concentration of the blank (0.071 copies/µL). Based on these findings, we determined the LOD to be >0.234 copies/µL (dilution 1/1,600) (Table 3 and Fig. S5). The LOD for *S. spallanzanii* was considered to be >0.380 copies/µL (dilution 1/6,400), and for *B. neritina* LOD was calculated as >0.536 copies/µL (dilution 1/12,800) (Table 3 and Fig. S5).

We considered the LOQ for *S. spallanzanii*, to be 0.698 copies/µL (dilution 1/3,200) (Table 3 and Fig. S6). Despite the CV of 36.5% observed for the *B. neritina* standard curve at dilution 1/1,600, we determined the LOQ for *B. neritina* to be 0.739 copies/µL at dilution 1/3,200. Dilutions below 1/3,200 consistently exhibited CV values exceeding 35% (Table 3 and Fig. S6). The *S. clava* assay had the most variability and we considered the LOQ for *S. clava*, to be 24 copies/µL (dilution 1/2.5) (Table 3 and Fig. S6).

## DISCUSSION

To the best of our knowledge, this study represents the first published evaluation and validation of direct-ddPCR capability to detect free-eDNA from targeted marine species through seawater sampling, marking the initial phase in the development of eDNA detection applications that could bypass the need for traditional sample processing steps.

### Feasibility of free-floating environmental DNA in marine environments

Despite the reduced susceptibility of ddPCR to inhibition, the high concentrations of salts and metal ions such as $K^+$, $Na^+$, and $Mg^2$ in saltwater samples can affect DNA stability and decrease the fidelity and enzymatic activity of DNA polymerase during the PCR reaction (*Davalieva & Efremov, 2010*; *Kuffel, Gray & Daeid, 2021*; *Lorenz, 2012*; *Mubarak et al., 2020*). Therefore, detecting free-eDNA directly from seawater samples is not a trivial task and requires *in vitro* optimization of the direct-ddPCR reaction, as was done in the present study.

Mitigating the inhibitory effects of salt was a critical step in achieving accurate detection and reducing the sample volume to 1 µL proved to be an effective approach (Fig. S3). Previous research has demonstrated that dilution can enhance reaction efficiency by reducing the concentration of inhibitors and has been used successfully as a method to
detect bacteria in drinking water (*Benson, Fode-Vaughan & Collins, 2004*; *Kokkoris et al., 2021*). However, it is important to acknowledge that using a 1 µL volume may not be optimal in highly dynamic marine environments, especially when targeting rare species or new incursions that may require larger water volumes to increase likelihood of detection (*Bowers et al., 2021*; *Diaz-Ferguson & Moyer, 2014*). To adapt this technique for field studies and reduce inhibition of salt further optimization of direct-ddPCR reactions may be necessary, such as using low salt ddPCR Supermix, testing alternative DNA polymerase, diluting the sample prior to testing in the reaction, and evaluating alternative polymer-buffer systems (*Hedman et al., 2010*; *Sidstedt et al., 2017*) (File S2). One example is the use of the multiplex master mix and buffering the water solution prior to sample collection, allowed for additional sample input. We found that using the ddPCR™ Multiplex Supermix and diluting the sample with buffers yielded a positive detection of free-eDNA during our in-vitro optimization trials. This is likely attributed to the replacement of the reduced volume of master mix with nuclease-free water and overall dilution, enabling an increase in the input sample volume up to 3 µL (File S2). Although even with optimization, the volumes will be lower compared to traditional methods, the benefit of this technology lies in its ability to allow for a massive increase in the number of continuous samples, enabling higher spatial-temporal coverage.

Additionally, we acknowledge that the distribution of free-eDNA within an ecosystem can be unpredictable and patchy (*Bowers et al., 2021*; *Eichmiller, Bajer & Sorensen, 2014*; *Harper et al., 2018*; *Itakura et al., 2020*). In this study, samples were only collected from the top ~5 cm of the tank, corresponding to the length of a P1000 tip. Consequently, certain free-eDNA molecules may have remained undetected despite aeration-induced mixing. Hence, researchers should tailor the testing of direct-ddPCR assays in the field to their specific environment and target species, considering factors such as detection probabilities and accordingly adjust sample dilution, volume, sampling depth, and sampling design.

## Effect of biomass and species on the detection of free-floating environmental DNA with droplet digital polymerase chain reaction

To explore the influence of species on the detectability of free-eDNA, this study focused on three morphologically and biologically distinct marine invertebrates: *S. spallanzanii*, *S. clava* and *B. neritina*. The results revealed significantly higher positive detection levels and free-eDNA concentrations for *S. spallanzanii* compared to both *B. neritina* and *S. clava* (Tables S1 and S2). These findings are consistent with the observations of *Wood et al. (2020)*, who also reported higher eDNA concentrations of *S. spallanzanii* compared with *S. clava* in laboratory-controlled conditions. The variations in detectability were attributed to anatomical differences between organisms, with *S. spallanzanii* potentially shedding more eDNA due to its fragile feeding tentacles, while *S. clava*'s tougher tunic may result in lower eDNA release (*Wood et al., 2020*). *Bugula neritina*, with its interconnected zooids enclosed within a calcified exoskeleton (*Keough & Chernoff, 1987*; *Trindade-Silva et al., 2010*), may exhibit lower release of free-eDNA compared to *S. spallanzanii* observed in this study. Given the observed detection differences among organisms in this study, we recommend that end-users tailor their eDNA detection

approach based on the target species, acknowledging that this method may not be viable for all species. To overcome interspecies detectability variations, researchers may consider increasing sample size, focusing sampling near the organisms' habitats, and accounting for the species' seasonal and life-history traits (*Burian et al., 2021*; *Mauvisseau et al., 2019b*; *Wood et al., 2020*).

The study also aimed to assess the potential impact of species biomass on free-eDNA detection. Previous studies have reported positive relationships between biomass abundance and eDNA concentrations (*Bradley et al., 2022*; *Doi et al., 2015*; *Everts et al., 2021*; *Lacoursière-Roussel et al., 2016*; *Rourke et al., 2022*; *Tillotson et al., 2018*). Although some recent studies have implied that eDNA may not be useful to infer abundance of some species (*Rourke et al., 2023*). In our analysis, we found significantly positive linear relationships between weight and free-eDNA concentration for *B. neritina* and a moderate positive linear relationship for *S. spallanzanii* (Fig. S7). However, *S. clava* showed a non-significant weak positive correlation, indicating a less pronounced relationship between weight and free-eDNA concentration (Fig. S7). Further analysis using a generalized linear mixed model (Table S4) revealed that weight alone did not have a significant effect on concentration. This suggests that weight alone may not be a reliable predictor of free-eDNA concentrations and that multiple factors, such as tank (intra-species variation) and species, which contribute to the variability in free-DNA direct-ddPCR data, should be considered when interpreting the results.

Variations in eDNA shedding have been observed among individuals, even when exposed to the same environment and exhibiting similar behaviour, with some studies reporting up to a 100-fold variation from the same fish under controlled conditions (*Rourke et al., 2022*). Factors such as stress and viability can influence interspecific variation in eDNA shedding, as seen in our study where animals, specifically *S. spallanzanii*, may have experienced stress during transportation and adaptation to the new aquarium conditions. Stress can create an imbalance between eDNA accumulation and decay (*Rourke et al., 2023*). Furthermore, the death of some organisms during the study could also have implications for eDNA shedding, with some models suggesting that shedding rates may increase after death (*Tillotson et al., 2018*).

Considerations should be given to the sensitivity and precision of species-specific ddPCR assays, as well as methodological aspects such as filtration and DNA extraction, as they significantly impact detection and recovery rates of eDNA (*Capo et al., 2021*; *Hinlo et al., 2017*; *Schweiss et al., 2019*). Assessing the performance of direct-ddPCR assays involves evaluating the LOD and LOQ, which reflect the assay's sensitivity and ability to accurately quantify low levels of target sequences (*Klymus et al., 2020*). In the present study, the LOD was similar across all three assays, but the LOQ for *S. clava* differed, suggesting the need for further optimization for routine point-of-need (PON) applications (Table 3). Evaluating sensitivity also requires considering the ddPCR chemistry employed, such as the hydrolysis probe (TaqMan, Carlsbad, CA, USA) and DNA binding dye (EvaGreen®, Taoyuan City, Taiwan). Both the EvaGreen assay (for *B. neritina*) and the duplex probe assay (for *S. spallanzanii* and *S. clava*) demonstrated comparable sensitivity in detecting free-eDNA, as indicated by the LOD values (Table 3). These findings align

with previous research that reported similar sensitivity for these ddPCR chemistries (*Falzone et al., 2020*; *McDermott et al., 2013*).

Our study focused on assessing the feasibility of detecting free-eDNA in seawater rather than directly comparing it with traditional methods. However, when comparing the ddPCR results from our study with previous research by *Wood et al. (2020)*, it becomes evident that traditional filtration and extraction methods may yield higher starting concentrations of *S. spallanzanii* and *S. clava* eDNA compared to the direct detection of free-eDNA. This disparity in methodology could potentially impact the detection rates and should be taken into account when designing a survey (*e.g.*, by adjusting the replication levels).

## Persistence of free-floating environmental DNA in seawater

Understanding the fate and persistence of free-eDNA is crucial for optimizing sampling strategies, improving detection accuracy, and interpreting findings (*Farrell, Whitmore & Duffy, 2021*; *Harrison, Sunday & Rogers, 2019*; *Zaiko et al., 2018*).

We observed that the free-eDNA signal for high biomass treatments of *S. spallanzanii* and *B. neritina* was detected up to 72 h, while that of *S. clava*—to only 24 h (Fig. 3). These findings contrast with a previous study by *Wood et al. (2020)*, who reported that the eDNA signal of *S. spallanzanii* declined below the detection limits of ddPCR within 35 h, while *S. clava* could still be detected up to 87 h. Although this suggests that free-eDNA may degrade at a slower rate than initially anticipated, these discrepancies also highlight the complexities and variability in eDNA degradation processes, which may be influenced by abiotic conditions, system setup, and inherent variability within the organisms (*e.g.*, stress, viability). It is also important to consider that free-eDNA may contain cellular contents released from damaged cells or lysed cell particles during high-temperature PCR cycles (*Shehadul Islam, Aryasomayajula & Selvaganapathy, 2017*). Therefore, further research is necessary to accurately classify and characterize free-eDNA using a combination of laboratory and in-field methods capable of capturing different size fractions of free-eDNA (*Jo et al., 2019*; *Moushomi et al., 2019*; *Turner et al., 2014*; *Wilcox et al., 2015*).

## Future directions

This study has demonstrated the feasibility of detecting targeted marine species directly from a water sample using direct-ddPCR, marking the initial step towards automated in-field technologies. While further work is necessary, such as expanding to other target plankton or nekton organisms, in-field experiments and development and validation of mobile ddPCR technology, there have already been promising strides toward in-field automation. Mobile devices capable of detecting targeted eDNA in field settings are actively being developed (*Hansen et al., 2020*; *Marx, 2015*; *Ponce, Arismendi & Thomas, 2021*; *Sepulveda et al., 2018*; *Thomas et al., 2020*). However, the current approach involves collecting and manipulating samples with a field-based DNA extraction kit followed by the mobile PCR device, reducing the practicality of such tools and introducing potential biases (*Marx, 2015*; *Nguyen et al., 2018*; *Sepulveda et al., 2018*; *Thomas et al., 2020*). Our results suggest that coupling direct PCR with ddPCR could potentially lead to the development of

automatic *in-situ* monitoring technology, mitigating some of these biases and enhancing sensitivity.

Along with targeted species analysis for eDNA, this study could help spark new ideas for *in-situ* assessment and screening of broader communities through the utilization of metabarcoding. However, for metabarcoding analysis, significantly larger volumes of water samples are needed for an accurate assessment of marine biodiversity (*Kawakami et al., 2023*; *McClenaghan et al., 2020*). In recent years, technologies such as robotic environmental sample processors (ESP) have been introduced with the ability to automate water sample collection and filtration, preserving the sample for further analysis (*Hendricks et al., 2023*; *Preston et al., 2023*; *Sepulveda et al., 2020*; *Truelove et al., 2022*), with some having the ability to filter large volumes (*Govindarajan et al., 2022*). Additionally, some second generations ESP can perform *in-situ* qPCR (*Hansen et al., 2020*). With the development of newer generations of ESP, it is easier to envision the collection and filtration of larger volumes and the ability to conduct direct PCR on filters (*Stojan et al., 2023*). Moreover, the success of mobile sequencing technologies such as Oxford Nanopore on marine eDNA samples (*Egeter et al., 2022*; *Truelove et al., 2019*), demonstrates the potential of coupling these technologies with ESP and direct PCR to create *in-situ* metabarcoding monitoring, although further ground-truthing of the technology is needed.

## CONCLUSIONS

To the best of our knowledge, this study marks the first published investigation of direct-ddPCR assays for the detection of free-eDNA from a seawater sample, circumventing the requirement for sample processing steps. The results demonstrate the feasibility of employing this technology for the detection of free-eDNA if salt inhibition is effectively addressed through assay optimization. The success of free-eDNA detection was influenced by the targeted species and their biomass. We were able to detect free-eDNA for up to 72 h following organisms' removal. These findings emphasize the importance of understanding the ecological characteristics of the targeted free-eDNA, such as dynamics of production/shedding, and longevity. It is crucial to develop assays that are customized for species and environments of interest. In summary, these encouraging results provide a foundation for the advancement and application of direct-ddPCR, acknowledging that further work is required if this technology is to be utilized in the field.

## ACKNOWLEDGEMENTS

We would like to thank Erin Bomati for her invaluable assistance in collecting and transporting *Sabella spallanzanii* samples. Additionally, we would like to express our gratitude to the staff at the Cawthron Institute, including Michael Scott, Daniel Cross, Juliette Bulter, Ian Saldanha, and Shaun Cunningham, for their support and valuable guidance throughout the aquarium experiment.

### Funding

This research was funded by the New Zealand Ministry of Business, Innovation and Employment funding (CAWX1904—a toolbox to underpin and enable tomorrow's marine biosecurity system). The funders had no role in study design, data collection and analysis, decision to publish, or preparation of the manuscript.

### Grant Disclosures

The following grant information was disclosed by the authors:
New Zealand Ministry of Business, Innovation and Employment funding: CAWX1904.

### Competing Interests

Xavier Pochon and Anastasija Zaiko are Academic Editors for PeerJ. Anastasija Zaiko is employed by Sequench Limited and Cody Youngbull is employed by Nucleic Sensing Systems, LLC.

### Author Contributions

- Michelle Scriver conceived and designed the experiments, performed the experiments, analyzed the data, prepared figures and/or tables, authored or reviewed drafts of the article, and approved the final draft.
- Ulla von Ammon conceived and designed the experiments, performed the experiments, authored or reviewed drafts of the article, and approved the final draft.
- Cody Youngbull conceived and designed the experiments, authored or reviewed drafts of the article, and approved the final draft.
- Xavier Pochon conceived and designed the experiments, authored or reviewed drafts of the article, and approved the final draft.
- Jo-Ann L. Stanton conceived and designed the experiments, authored or reviewed drafts of the article, and approved the final draft.
- Neil J. Gemmell conceived and designed the experiments, authored or reviewed drafts of the article, and approved the final draft.
- Anastasija Zaiko conceived and designed the experiments, authored or reviewed drafts of the article, and approved the final draft.

### Field Study Permissions

The following information was supplied relating to field study approvals (*i.e.*, approving body and any reference numbers):

All samples were collected under the specifications of permission to handle organisms under Sections 52 and 53 of the Biosecurity Act 1993 (Notice No. MPI 111) from the New Zealand government agency Ministry for Primary Industries.

### Data Availability

The raw data, tables and figures are available in Supplemental Files.

## Supplemental Information

Supplemental information for this article can be found online at http://dx.doi.org/10.7717/peerj.16969#supplemental-information.

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
