# Peer review of "Drop it all: extraction-free detection of targeted marine species through optimized direct droplet digital PCR"

_PeerJ, doi:10.7717/peerj.16969_

## Round 0.1 · original submission · Minor Revisions

The manuscript is well-written and clearly explains the method and application of direct amplification of free-floating eDNA using ddPCR in detecting invasive marine species.

I agree with the second reviewer who stressed that this method could be generalized for detecting species of interest, not only invasive (eg, endangered species). This work used benthic sessile species as a case study. Would you extend its use to plankton and nekton organisms?

Further issues raised by the reviewers regarding overcoming the variability in the field, comparison with qPCR and other methods, and adding more literature should be addressed.

Reviewer 1 ·

Basic reporting

Scriver et al. report on experiments testing the direct amplification of free-floating DNA for three marine invasive species. They show that eDNA is composed of extracellular DNA that can be detected when the species were present at high abundance and last for up to 1-3 days after the species have been removed. This study represents an innovative addition to the field of eDNA as it provides with new tools for faster species detection. I found it insightful to see that biomass was not a reliable predictor of free-eDNA concentrations, as other experimental studies (relying on eDNA extraction) generally describe a relationship between those two.

It is interesting to see such big interspecific differences in detectability. I would like to see one or two sentences on how could researchers overcome these differences on the field - maybe by collecting more replicates, or sampling closer to the species' habitat?

I would also like to see a couple of sentences on "future directions", for example, how could this method be extrapolated to eDNA metabarcoding?

Finally, could the authors comment on how could the results differ if carried out using conventional qPCR?

I am looking forward to seeing this ms published on PeerJ.

Experimental design

The research question was framed appropriately and the experimental design is robust.

Validity of the findings

The findings are novel and impactful. The underlying data has been provided and is statistically sound and controlled. The conclusions are well stated.

·

Basic reporting

no comment

Experimental design

no comment

Validity of the findings

no comment

Additional comments

This study details an interesting and important attempt in the innovative world of eDNA.
The paper is well written and explained.
While I had only minor comments that can be seen in the attached draft, I would like to raise a specific conceptual comment that I also noted in the draft about the focus on NIS in this study.
This study is clearly a proof of concept for a new methodology, and its relation to invasive biology is merely suggestive and not strongly related with the methods itself.
With this in mind, I encourage the authors to decrease a little the focus of NIS to this study.

Also, I found citations in the body of the text that were missing from the reference list.
Please carefully check me here.

Best

---

## Round 0.2 · accepted · Accept

Well done. All comments were properly addressed. I am looking forward to seeing this paper online.